# Six-Steps Process of Structural Assessment of Heritage Timber Structures: Definition Based on the State of the Art

**Elena Perria** * 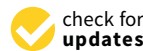 **and Mike Sieder**

Institute of Building Construction and Timber Structures (iBHolz), Faculty of Architecture, Civil Engineering and Environmental Sciences, Technische Universität Braunschweig, 38106 Braunschweig, Germany; m.sieder@tu-braunschweig.de
* Correspondence: e.perria@tu-braunschweig.de

**Abstract:** Each construction material deals with specific mechanical properties, their distribution, damage mechanisms, and degradation processes. Therefore, each material requires a particular assessment approach in order to derive a reliable description of the residual performance of the structure, correctly remove the cause of damage, and proceed with the correct design of interventions. The aims of this paper are, first, the definition of a process of assessment and retrofitting of existing timber structures, both for engineered and heritage/traditional timber structures, and second, a comparison between the defined assessment process and its contents, and the content of existing guidelines, codes, and standards. In order to gain a definition of the process of assessment and retrofitting of existing timber structures, the content of scientific papers and articles was analyzed, and on this basis, an assessment process with six main steps and three milestones was developed. The content of the guidelines, codes and standards was afterwards analyzed basing on this six-steps assessment process. From a comparison among the current literature, guidelines, codes, and standards, interesting results emerged that gave us a picture of the European knowledge and interests on the assessment of existing timber structures. Not only agreement, but also discrepancies, variances, and incongruities were identified as possible topics for future research.

**Keywords:** structural assessment; guidelines; heritage structures; traditional structures; timber; standards

## 1. Introduction

The condition survey and diagnosis of structural elements are the first steps for the assessment of existing timber structures in order to proceed with the planning of interventions for continuing or change of use. Existing buildings offer a lot of information on their substance that have to be ordered and understood. Apart from the knowledge of the specific heritage building, what is needed is a priori know-how about historical-aesthetic, architectonical-technical, as well as, for each construction material, engineering-technical aspects like assessment, monitoring and conservation methods. Hence, this process needs the contribution of different experts (architects, engineers, historians, archeologists, restaurateurs, etc.) to generate a multidisciplinary report on a structure's condition.

For the preparation of a report on the state of conservation of an existing building, two main aspects are required: theoretical and practical knowledge. On one side, there are formal contents: survey methods and processes, restauration theories, and inspection and documentation forms. On the other side, there is a huge amount of practical information collectable on the existing substance. Not only the characterization of materials and their decay have to be documented with the aim of elimination of causes of decay, structural analysis, and design of interventions, but also knowledge



about the structural system and their formal and stylistic peculiarities form part of the information to be gathered on-site, with the aim of correct preservation of the heritage value.

The final aim of this paper is the definition of a common process for the structural assessment of timber heritage buildings with homogeneous "terms" and "definitions" at the European level. The definition of the process of structural assessment of heritage timber structures derives from the analysis and comparison of good practice reported in scientific papers and articles that deal with the structural retrofitting of timber buildings. What is more, this paper aims to bring forward the proposed methodology and compare it with the content of current norms, standards, and guidelines. From this comparison, the paper will highlight the overlapping and the missing topics both in scientific papers and in codes and guidelines, in order to determine discrepancies, variances, incongruities, and opportunities for future research.

The paper is divided in two main sections: "Review of papers" and "Review of existing codes and guidelines". The first section contains the analysis of selected papers and articles that deal with the procedure of a condition state survey for both traditional and engineered timber structures. Based on a résumé of the procedures proposed by these documents, the process of assessment and retrofitting of existing timber structures was resumed into six main steps and three milestones. The second section contains a review of existing codes and standards on structural assessment with particular attention paid to the ones that consider timber as main material, and the ones that deal with heritage/traditional timber structures. The review process includes many other codes and standards referenced in the mentioned standards. Based on this six-step assessment process described in the first section, the content of these existing codes and standards are analyzed. The comparison between the state of the art of literature, guidelines, codes, and standards is held with the critical eye of the expert that uses codes for the conservation of modern and ancient timber buildings. The standards are evaluated as a tool for the engineer that wants to methodically proceed with the analysis of the building.

## 2. Review of Papers

### 2.1. Introduction

The number of papers and articles that deal with the description of the condition state of existing structures is increasing. Most of these documents mainly focus on the assessment of the general structure of any construction material. Nevertheless, due to the increasing importance of wood as a new and rediscovered construction material, there is also an increasing amount of literature about the assessment and retrofitting of timber constructions, among which there is also an increasing number of papers deal with the assessment of heritage/traditional timber structures.

In this paper, papers and articles that deal with the topics of the assessment process of heritage/traditional and engineered timber buildings are analyzed. From the analysis of selected papers and literature that consider the survey conditions of existing heritage/traditional and engineered timber structures, the definition of the single steps and milestones of the process of assessment and retrofitting of existing timber structures was extrapolated. The six steps and three milestones of the process of assessment and retrofitting of existing timber structures are described with a brief description of their contents in the Figure 1. The proposed assessment process will be described by referencing the single literature references in Section 2.2.

Process of assessment and retrofitting of existing timber structures:

Step 1: Desk survey: collection of existing information on the structure.

Step 2: Preliminary survey: description of the actual structure.

Milestone 1: Preliminary report.

Step 3: Decay and check survey: grading of structural members.

Step 4: Structural analysis: evaluation of static-constructive safety and serviceability of building components.

Milestone 2: Diagnostic Report.

Step 5: Analysis of special requirements: dealing with special requirements.

Step 6: Design of interventions: design of refurbishment, repair, or strengthening interventions.

Milestone 3: Executive report.

**STEP 1:**
**Desk Survey**

- study of (historical) documents and other evidence
- construction year
- planned static system
- original construction type
- documented refurbishments / change of use
- building load history
- determination of future use of the building

**STEP 2:**
**Preliminary**
**Survey**

- geometric survey:
  - identification of the general structural system
  - identification of load-bearing and non-load bearing structural elements
  - identification of structural changes / differences compared to the original construction documentation
  - identification of visible deformations
  - identification of wooden species and intended form (structural timber, glued laminated timber, cross laminated timber, etc....)
  - survey of cross-sections of structural elements
  - identification of deformations and displacements/imperfections
- determination of the extent of visible mechanical damage/decay/alterations
- decision on immediate safety interventions

**Milestone M1: Preliminary Report**

**STEP 3:**
**Decay &**
**Check Survey**

- determination of deformations (impact on load capacity / effects on serviceability)
- determination of modulus of elasticity (E-Modulus) of main members
- grading of timber elements (determination of mechanical properties)
- assessment of remaining cross-section of structural members
- determination of wood moisture content (MC)
- determination of cracks / mechanical damage/ (static critical / uncritical)
- determination of the real extent and depth of decay / alterations and possible causes
- assessment of (remaining) timber properties (impact on load capacity)
- assessment of timber-timber connections (intact load transfer / no contact)

**STEP 4:**
**Structural**
**Analysis**

- determination of distribution of the forces in the structure
- assumptions about the actual static system, included connections
- load assumption / design loads
- evidence of load-carrying capacity of the structural system
- punctual evidence of load-carrying capacity of connections
- evidence of buckling (compressed elements) and torsional buckling (bent elements)
- estimate/determination of future working load due to the use of the building
- verification of serviceability limit state
- verification of structural safety and reliability

**Milestone M2: Diagnostic Report**

**STEP 5:**
**Special**
**Requirements**

- design requirements (building regulation, monument preservation, etc...)
- requirements on energy efficiency
- wood preservation
- fire protection
- sound insulation

**STEP 6:**
**Design of**
**Interventions**

- local replacement of connections or structural elements with same material (no changes in the existing structural scheme)
- bracing measures, reinforcement or replacement of existing elements with new ones (upgrading, changes in the existing structural scheme)
- addition of supporting structures / constructions
- verification of load-carrying capacity of the structure (same material and same structure)
  - calculation according to old standards at the time of construction
- verification of load-carrying capacity of the existing structure with increased loads or reinforced / upgraded (changes in the existing structural scheme)
  - calculation according to current calculation rules

**Milestone M3: Executive Report**

**Figure 1.** Six-steps assessment process of timber structures.

## 2.2. Reviewed Papers

The first step of the assessment process is the desk survey. The desk survey is proposed for the assessment of both engineered and traditional timber structures. In Blaß [1] and Dietsch, Koehler [2], the focus is on large span structures. Here, the collection of documents about the planned structural system, its static verification, as well as information pertaining to the original structure, reparations, structural changes, and finally, determination of future use is suggested. Dietsch, Koehler [2] refer to the approach already proposed by Diamantidis [3]. Lißner, Rug [4] complement this step for traditional timber structures with the introduction of the documentation on building history, and Cruz, Yeomans, et al. [5] with the research on historical aspects of the original constructional art, and building's loading history.

The second step of the assessment process is the preliminary survey. It aims at the description of the general actual conditions of the structure. This step forms the core of the assessment process in all the analyzed literature references. For the general timber structure, this step is described as a "re-design" of the existing structure with the goal of understanding the object [6]. Lißner, Rug [7] additionally describe all single aspects to take into account during the assessment of the existing construction, focusing not only on true-to-deformation geometric survey, but also on the importance of a detailed analysis of the cross-sections, connections, bracing system, wooden species, and identification of the original load-carrying structure. Rug, Stützer, et al. [8] remark on the importance of a visual damage survey and detailed analysis of woodworking connections. In fact, the latter, together with the extent of mechanical damage or alterations could be related with safety issues. Furthermore, in D'ayala, Branco, et al. [9], a first draft of "vulnerability assessment forms" for the assessment of the seismic vulnerability of timber roofs has been developed. The template is divided in two main parts, the first one focuses on the record of the structure and all structural elements, and the second one on the record of damages, defects, their assumed cause(s), and structural consequence(s). Riggio, D'Ayala, et al. [10] reviewed the second assessment tool, among others.

All the pieces of information collected during these first two steps must be reported in a first milestone (M1), the preliminary report, which describes the tasks already performed and points out any additional detailed or instrumental survey necessary to be carried out in the next steps. The Milestone 1 was found in references [2,3,5,6,9].

Step 3 of the assessment process regards the decay and check survey. During this step, the previous visual damage survey will be detailed in a measured inspection with a collection of quantitative information on the causes and extent of decay in the constructional elements. Furthermore, characteristics, imperfections, and any useful information for grading of timber elements must be performed [6]. The criteria for timber grading are extensively described by Lißner, Rug [7]. The inspection tools are described by Dietsch, Koehler [2] and in the Wissenschaftlich-Technische Arbeitsgemeinschaft für Bauwerkserhaltung und Denkmalpflege (WTA) document [11]. The grading classes assigned for the elements can be "translated" in strength classes by taking into account the strength classes proposed in EN 1912:2012 [12]. Lißner, Rug [7] clarify that the existing grading rules are made for sawn timber. Therefore, the final grading for heritage or traditional timber must differ from the one proposed in EN 1912:2012 [12], and the inspection should concentrate on the points of maximum stress/strain in the structure. Also, in Cruz, Yeomans, et al. [5], some alternative criteria for the timber grading according to different stresses in structural members are offered. Furthermore, focusing on heritage timber structures, Cruz, Yeomans, et al. [5] highlight the importance of the dating of wood that can be reached by means of dendrochronological investigations.

Once the geometric/deformation analysis and the detailed grading of timber has been completed the structural analysis in the step 4 can be approached. The aim of the static analysis is the evaluation of the static-constructive safety and serviceability of the building components [7]. All literature references concur that a good structural analysis relies on the correct collection of pieces of information during the previous steps. Dietsch, Koehler [2], based on Diamantidis [3], further suggests the updating of the collected data on material properties with the formulas proposed in the Probabilistic Model

Code [13]. Once the material properties have been fixed, the determination of the distribution of forces and a realistic assumption of the static system should be modelled. The WTA document [11] and Holzer [14] give suggestions for the modelling of historical timber members and connections. The load assumptions should be done according with the current norms [11]. Finally, for the evidence of ultimate limit state and serviceability limit state, in case of addition of new structural members or changes in the original structural system, the calculation should be performed according with current norms. Alternatively, in case of the replacement of existing timber elements, the verification can be calculated according to the norm valid at the time of the construction [15] or a local verification of the repair connection according with current norms can be performed. The Probabilistic Model Code [13] also proposes an alternative to the verification of ultimate limit state and serviceability limit state relating to the actual environmental conditions of the existing building.

A second milestone (M2) with the name of diagnostic report is suggested by Cruz, Yeomans, et al. [5]. Here, the condition of the structure and causes of distress with proposals for remedial measures should be reported. This report may imply the structural analysis considering the data gathered in the two latter performed assessment levels. Dietsch, Koehler [2] and Diamantidis [3] also propose this second milestone.

In many cases of renovation of historical buildings for current use, the approach to the conservation of heritage/traditional structures joins other special requirements connected to modern concepts of comfort and safety. These issues introduce step 5 of the conservation process, namely the analysis of special requirements. Lißner, Rug [7] show in a flow chart the special requirements, namely design, wood preservation, energy saving and moisture proofing, acoustic insulation, fire protection, and, of course, design requirements strictly connected with the cultural heritage preservation. These requirements that could differ for content in each country are described in detail for Germany in Lißner, Rug, et al. [16].

Step 6, the design of interventions, is the final step of the proposed process in which the refurbishment/repair/strengthening interventions on the structure are described. The interventions have to be distinguished for load-bearing and non-load-bearing elements [8]. In fact, the latter would be performed only to improve the comfort in the building (step 5), but should be correctly chosen in order to avoid future structural damage. Throughout the analyzed literature, there exists the unanimous idea that any historical structure is unique, and the interventions must be always tailor-made for the special case, by specialists that already have expertise in the field of historical timber structures. In Rug, Stützer, et al. [8], some refurbishment/repair/strengthening methods and examples for half-timber structures and roofs are suggested. Further examples are contained in the didactic book [4] in which not only examples of repair and reinforcement for the timber structure, but also of fire protection, sound insulation and energy efficiency refurbishment, strictly connected with step 5, are offered. Furthermore, D'Ayala, Branco et al. [9] present a "tool for reinforcement selection", created with the aim to assist engineers and architects to select the most suitable reinforcement solution among a range of alternatives. Here, solutions for typical problems in timber elements are sorted and ranked according to a range of "damage" criteria. A prototype tool, namely the TimberSave, has been developed as an App for Android platforms. Step 6 has been defined as "detailed design of repairs" by Cruz, Yeomans, et al. [5] and as "constructional measures" by Lißner, Rug [7].

Last milestone (M3) is the executive report. This is a detailed report in which the chosen refurbishment/repair/strengthening interventions, executive drawings, and execution works are summarized [2–4,7,16]. This report concludes the process of structural assessment for existing timber structures.

An overview of main topics of selected papers and their contents for the definition of the steps and milestones according to Figure 1 are given in the Table 1.

**Table 1.** Content of selected papers. Each paper is reviewed considering the identified steps (S) and milestones (M) according to the *Six-steps assessment process* of Figure 1. For each paper, the discussed contents are reported.

| Reference Nr. | First Author/Title/Publication Year | Timber Topic | | Content Related to the Assessment Process | | | | | | | | |
|---|---|---|---|---|---|---|---|---|---|---|---|---|
| | | Heritage/Traditional | Engineered | Step 1 | Step 2 | M1 | Step 3 | Step 4 | M2 | Step 5 | Step 6 | M3 |
| [1] | S.H. e.V./Leitfaden zu einer ersten Begutachtung von Hallentragwerken aus Holz/2018 | | X | 1 | 2 | | 3 | | | | | |
| [2] | Dietsch/Assessment of Timber Structures. Report of the COST Action E55 "Modelling of the Performance of Timber Structures"/2010 | | X | 1 | | M1 | 3 | 4 | M2 | | | M3 |
| [3] | Diamantidis/Probabilistic Assessment of Existing Structures/2001 | | X | 1 | | M1 | | 4 | M2 | | | M3 |
| [4] | Lißner/Holzbausanierung—Grundlagen und Praxis der sicheren Ausführung/2000 | X | | 1 | | | | | | | 6 | M3 |
| [5] | Cruz/Guidelines for on-site assessment of historic timber structures/2015 | X | | 1 | 2 | M1 | 3 | | M2 | | 6 | |
| [6] | Macchioni/Review of Codes and Standards/2010 | | X | | 2 | M1 | 3 | | | | | |
| [7] | Lißner/Ergänzende Erläuterungen für Bauen im Bestand/2005 | X | | | 2 | | 3 | 4 | | 5 | 6 | M3 |
| [8] | Rug/Erneuerung von Fachwerkbauten/2004 | X | | | 2 | | | | | 5 | 6 | |
| [12] | CEN/EN 1912:2012. Structural Timber—Strength classes—Assignment of visual grades and species/2012 | | X | | | | 3 | | | | | |
| [13] | JCSS/Probabilistic Model Code, Part III, Timber/2001 | | X | | | | | 4 | | | | |
| [11] | WTA/Merkblatt E-7-2. Historische Holzkonstruktionen: Zustandsermittlung und Beurteilung der Tragfähigkeit geschädigter und verformter Holzkonstruktionen/2018 | X | | 1 | 2 | | 3 | 4 | | | | |
| [14] | Holzer/Statische Beurteilung historischer Tragwerke Band 2, Holzkonstruktionen/2005 | X | | | | | | 4 | | | | |
| [15] | ARGEBAU/Hinweise und Beispiele zum Vorgehen beim Nachweis der Standsicherheit beim Bauen im Bestand/2008 | | | | | | | 4 | | | | |
| [16] | Lißner/Modernisierung von Altbauten/2001 | X | | | | | | | | 5 | 6 | M3 |
| [9] | D'Ayala/Assessment, reinforcement and monitoring of timber structures: COST action FP1101/2014. | X | | | 2 | M1 | 3 | | | | 6 | |

## 3. Review on Existing Codes, Guidelines and Standards

### 3.1. Introduction

Codes, guidelines, and standards are written with the general purpose to guide experts in the correct development of their professional tasks. Codes, guidelines, and standards that deal with the conservation of heritage structures indirectly influence the approach to the survey and the retrofitting. Therefore, it is of basic importance that they offer adequate tools for the specific construction material, in order to lead the expert in the correct assessment process.

In this section, codes, guidelines and standards that focus on the assessment process of existing structures based on the six-step assessment process defined in the Section 2 are analyzed. The analysis of these documents is resumed in the Tables 2 and 3. The analyzed codes, guidelines and standards, differ for topic – intended as construction material (timber or other material)—and for deepness and specificity of content. In particular for each of the documents is indicated:

- if the approach to the conservation is for any kind of construction material or specific for timber (respectively "A" or "T" under "Material")
- if the approach to the conservation is for heritage or non-heritage structures (respectively "H" or "N" under "Material" and "Timber")
- if the contents are qualitative or quantitative
- the content of each of the steps of the proposed assessment process.

Among the analyzed documents there are, in chronological order, the Italian standards UNI 11119:2004 [17] and UNI 11138:2004 [18], some International Council on Monuments and Sites (ICOMOS) charters and guidelines (publication years: 1999, 2003, 2017), the ISO 13822:2010 [19] and its Annex I - Heritage Structures, and the Swiss norm series SIA 269/5:2011 [20]. Furthermore, the current Eurocode 5 with German national Annex [21], the proposal for the new European Technical Rules for Assessment and Retrofitting of Existing Structures [22], the JCSS Probabilistic Model Code, section Timber Structure [13], and the norm EN 17121:2019 [23] were analyzed.

### 3.2. ICOMOS/ISCARSAH Charters and Guidelines

The scope of the International Scientific Committee for Analysis and Restoration of Structures of Architectural Heritage (ISCARSAH) is "integrating the contribution of structural engineering into conservation knowledge, so that a full understanding of [ … ] behavior and material [ … ] become an intrinsic part of conservation practice".

The ICOMOS document [24], presenting principles for the preservation of historic timber structures, points out in a general descriptive way most important factors to pay attention to when dealing with a heritage timber structure. Particular attention is given to the phases of documentation, diagnosis (step 2 and step 3) and of design of interventions (step 6). The focus is on a reversible intervention, distinguishable from the existing structure, performed with durable materials, compatible with wood, and that follows the principle of minimum intervention. The use of wood as main repair material is strongly suggested.

The document *Principles for the analysis, conservation and structural restoration of architectural heritage* [25] describes qualitative general assessment criteria valid for all heritage construction materials. The document gives qualitative indications for the diagnosis (Step 3) and structural model (Step 4), and concludes with very general concept of "proposal for interventions" (Step 6).

**Table 2.** Résumé of the content of the analyzed standards—Step 1 to Step 3.

| Material | | Timber | | Norm/Guideline | Step 1 | | Step 2 | | Step 3 | |
| H | N | H | N | | Qualitative | Quantitative | Qualitative | Quantitative | Qualitative | Quantitative |
|---|---|---|---|---|---|---|---|---|---|---|
| T | | | x | ICOMOS 2017 | Point 21: definition of the original structural principles | | Point 1: record the structural and architectonical condition of the structure | | Point 2: accurate diagnosis of the physical conditions using NDT testing | |
| A | | | | ICO-MOS 2003 | Section 2 | | | | Section 2 | |
| A, T | | | x | ICO-MOS 2014 | Sections 2.1 and 2.2—"Pre-survey" | | Section 2.3 | | Sections 2.3 and 2.4.—Improvement by monitoring phase | |
| A | A | | | ISO 13822:2010 | Section 4.5.1.—Study of documents and other evidence | | Section 4.5.—"Preliminary assessment" | | Sections 4.6.2. and 4.6.3.; Section 5 —method of assessment; Annex D Assessment of actual mechanical properties | Annex C—Assessment in terms of probabilistic methods and estimated reliability |
| T | | | x | ISO 13822: 2010-ANNEX I | I.4.5.2.to I.4.5.4—Historical report, detailed documental search and review, heritage record | | Section I.4.4—Importance of samples collection according with cultural heritage management authorities | | Sections I.4.5. and I.4.5.5. to I.4.5.8—samples collection according with cultural heritage management authorities | |
| T | | | x | UNI ISO 11119:2004 | | | Section 7.1 to 7.4 —Identify wooden specie and local/global moisture content | Reference: UNI 11118, UNI EN 335 | Section 7.5 to 7.6—Grading of wood (growth characteristics) | Table 1, 2—Grading of structural elements; Appendix A—Common wooden species and visual grading classes |
| T | | | x | UNI ISO 11138:2004 | Section 4.2—"Historical survey" | | Section 4.3 and 4.4—"Material characterization" and "geometric survey" | Reference: UNI 11118, UNI 11119 and UNI EN 335 | Section 4.5—"Characterization of the decay" | |
| T | | x | x | SIA 269/5:2011 | | | Section 5—Structural survey | | Annex C.2—Procedures and tools for the assessment; Section 6—Use classes. Reference: SIA 265:2003, SIA EN 335-2 | Annex C3—Common decay in wooden connections; Annex D—tables from EMPA/LIGNUM |
| T | | x | x | DIN EN 1995-1-1/ NA: 2013-08 | | | | | | |
| A | A | | | NEW EUROPEAN TECHNICAL RULES FOR [ … ] EXISTING STRUCTURES | | | Part III, Section 3.2—"Generic procedure"; Section 3.3—"Preliminary assessment". | | Part III, Section 3.4 "detailed assessment"; Section 4 "investigation and updating information"; Section 4.3 "Material properties" (Ref: EN 1991 to 1999), Section 4.3.2 "Sampling and material testing"; Section 4.4 "Geometrical properties". | |
| T | | | x | EN 17121:2019 | Section 4—Preliminary assessment (including 4.2 "desk survey" and 4.3 "historical analysis") | | Section 4—Preliminary assessment including 4.5 "preliminary visual survey" and "measured survey" | Annex A—Tools for inspection briefly described | Section 5.5—Principles of timber grading; Section 5.7—Assessment of timber joints characteristics | Common joint's failure modes, Reference: EN 138183-2 and 3, EN 14081-1:2016, Annex A. |

**Table 3.** Résumé of the content of the analyzed standards—Step 4 to Step 6.

| Material H | N | Timber H | N | Norm/Guideline | Step 4 Qualitative | Quantitative | Step 5 Qualitative | Quantitative | Step 6 Qualitative | Quantitative |
|---|---|---|---|---|---|---|---|---|---|---|
| T | | | x | ICOMOS 2017 | x | | x | | x | |
| | | | | | Point 19: performance over the centuries and structural changes; Point 20: Adequate structural calculation | | Points 23–25: taking care of modernization methods in order to take care of the pre-existing heritage structure. | | Point 10–12: minimal, reversible intervention with traditional techniques; 21: basing interventions on future use; 23—Importance of compatibility and durability of new materials; 29–31: Monitoring | |
| A | | | | ICO-MOS 2003 | x | | | | x | |
| | | | | | Section 2 | | | | General concept of proposal for interventions | |
| A, T | | | x | ICO-MOS 2014 | x | | | x | | |
| | | | | | Section 3 | | 3.4—actions on structures and materials | | | |
| A | A | | | ISO 13822:2010 | x | x | | | x | |
| | | | | | Section 4.6.3.–4.6.6.; Section 6 and 7—Generation of a suitable structural model. Reference: ISO 2394:1915 | Annex E: Time-dependent and Annex F: Target reliability level | | | Sections 9 and 10; Annex H (Methods of Upgrading) | |
| T | | | x | ISO 13822: 2010—ANNEX I | x | | x | | x | |
| | | | | | Sections I.4.5.11 and I.6—care for development of structural model; Section 4.6.5 and 6.1 – Requirements on models. | | Section I.2—Fundamentals for the assessment of heritage structures, in general Annex I | | Section I.9—Minimal intervention, incremental approach, removable measures. | |
| T | | | x | UNI ISO 11119: 2004 | x | x | x | | | |
| | | | | | Reference: UNI EN 518 (replaced by EN 14081) | Reference: UNI EN 1995-1-1 | Section 6—Limits of the norm | | | |
| T | | | x | UNI ISO 11138:2004 | x | x | x | | x | x |
| | | | | | Section 4.6 and 4.7—"Structural analysis" and "presentation of results" | Reference: UNI EN 1995-1-1, DIN EN 1991-1-1, DIN EN 1998-1 | Introduction; Section 0—Special compatibility requirements of the heritage | | Section 5—Description of criteria for design of interventions. Requirements for preliminary, definitive and executive project. | Reference: UNI 8662-2:1988 |
| T | | x | x | SIA 269/5:2011 | x | x | | | x | |
| | | | | | Section 4—Evidence of load-carrying capacity, serviceability and connections | Annex B—Definition of load-carrying capacity of wood-wood connections with wooden fasteners; Annex A—Ref: SIA164:1953 and SIA 164:1992 | | | Reference: EMPA/LIGNUM-Guideline, SIA 269, SIA 269/1 and SIA 265. | |
| T | | x | x | DIN EN 1995-1-1/ NA: 2013-08 | | x | | | | |
| | | | | | Section NA 12.1—Double step joint; Section NA 12.2—Notched connect; Section NA 12.3 Nailed connection. | | | | | |
| A | A | | | NEW EUROPEAN TECHNICAL RULES FOR [ … ] EXISTING STRUCTURES | x | x | x | | x | x |
| | | | | | Part III, Section 3.4.6—Structural analysis; Section 4.2—"Actions" (Reference: EN 1990), Section 5—"Structural analysis and verification" | Reference: ISO 13822:2010, ISO 2394:2013, JCSS PROBABILISTIC MODEL CODE | Part III, Section 1.1—Document does "not cover assessment related to special performance requirements of the client related to property protection". | | Part III, Section 6—"Interventions"; Section 6.3—"Survey and monitoring maintenance" | Reference: *S&P reports* for EN 1992 to EN 1999. |
| T | | | x | EN 17121:2019 | x | | x | | | |
| | | | | | Section 4.6—Structural analysis; Section 5.7—Tool for understanding /modeling of joints. | | Introduction | | | |

*Recommendations for the analysis, conservation and structural restoration of architectural heritage* [26] provide qualitative indications for all the steps of the assessment process, except for the design of interventions. Important focus is given to the knowledge of the structure, that requires information on its conception, historic and cultural significance, on its original building materials and construction techniques, on the processes of decay and damage, on changes and any previous structural interventions (step 1). The section related to step 4 does not provide any numerical solution about how to calculate actions on structures and materials but gives some qualitative advice about the simulation of the structural behavior. For example, mechanical and physical actions may be simulated separately; this latter action may be introduced in the calculation by reducing mechanical properties of the correspondent construction material.

The purpose of the *Principles for the conservation of wooden built heritage* [27] adopted by the 19th ICOMOS General Assembly, New Delhi, India, 15 December 2017 is to update the contents of the ICOMOS document [24]. This document defines the basic principles and practices applicable in the widest variety of cases for the protection and conservation of the wooden built heritage with respect to its cultural significance. As in the original document, particular attention is given to the phases of documentation, diagnosis (step 2 and step 3), and of design of interventions (step 6). Step 2 and step 3 point out first, the importance of recording the structural and architectonical condition of the structure; afterwards, the importance of an accurate diagnosis of the physical conditions using non-destructive testing (NDT) is remarked. Before proceeding with the design of interventions (step 6), the document points out the importance to justify any intervention taking into account the existing structural principles and basing on future use of the building, connecting step 6 with step 1. The principles of intervention are the same of the first document: the priority should be given to traditional techniques, compatible and reversible materials. Some additional notes on step 6 are given: on one side, if a structure has shown a satisfactory performance, and if the use, actual conditions and loading regime remained unchanged over the centuries, the structure can be adequately retrofitted by simply repairing damage and failure with the same material and technique; on the other side, if alterations have been made to the structure, or any proposed change of use is planned, the residual load-bearing strength should be estimated by structural analysis before considering the introduction of any further reinforcement. Finally, this guideline offers some qualitative remarks on step 5, the modernization of the structure, highlighting the importance of the correct choice of the modernization methods and materials in order to protect the pre-existing heritage structure.

### 3.3. The European Standard ISO 13822:2010

The European International Standard ISO 13822:2010 [19], provides general information and methodological background on how to perform the assessment of any type of existing structure (buildings, bridges, industrial structures, etc.) that was originally designed, analyzed, and specified based on accepted engineering principles and/or design rules, as well as structures constructed based on good workmanship, historic experience and accepted professional practice. The standard covers all steps of the assessment process, except Step 5 dedicated to special requirements. The standard suggests an assessment in accordance with the ISO 2394:2015 [28] in which the assessment procedure bases on the principles of structural reliability and consequences of failure proposed in the Probabilistic Model Code [13]. The assessment process has to be adjusted to the objectives defined in the Probabilistic Model Code [13] based on "performance levels" connected with the strategic importance of the building in terms of safety and serviceability. The general assessment methodology is proposed in Annexes A and B in the form of flowcharts, and in Annex G as a "test report format". The norm is also applicable to heritage structures providing additional considerations shown in Annex I "Heritage Structures". Here, the same general procedure is proposed for the assessment, but special requirements (step 5) connected with heritage buildings are taken into account. The concept of reliability contained in the Annex F is here transferred for the specific requirements of heritage structures. Finally, a particular accuracy for the static model is required because of "uncertainties in data, damage and deformation,

limited knowledge of ancient structural system and variability of material properties". The "test report format" proposed in the Annex G of the main section is implemented with the points in Section 1.10 of the standard.

The ISO 13822:2010 [19] was updated in a set of Czech national annexes named CSN ISO 13822:2010 [29]. Presently, the revised version of this standard is implemented in the CSN 730038:2014 [30], where in Chapter 8 the "assessment of existing timber and timber concrete structures" is debated. This document has not been analyzed for this paper.

*3.4. National Standards UNI, SIA and DIN*

Europe, Italy, and Switzerland are contributing with standards at national level about the monitoring and evaluation specific for wooden artifacts. The two Italian standards UNI 11119:2004 [17] and UNI 11138:2004 [18] refer exclusively to historical timber built heritage. The Swiss norm series SIA 269:2011 [31] contains standards for the maintenance and reassessment of existing structures, including specific adaptive rules for actions on existing timber structures in SIA 269/5:2011 [20]. Furthermore, Czech Republic and Germany contribute with two national annexes to the European standards with CSN ISO 13822:2010 [29] and DIN EN 1995–1–1/NA:2013–08 [21] respectively.

The UNI 11119:2004 [17] establishes, on one side, qualitative objectives, procedures and requirements for the assessment of the state of conservation of heritage timber buildings. In step 2 it refers to UNI EN 335:2013 [32] and in step 6 to CEN EN 1995-1-1:2004 [33] for more quantitative assessment. On the other side, for step 6, a quantitative direct assessment and grading of structural elements is proposed. In Appendix A, based on visual grading classes, strength and stiffness values for the most common wooden species for the calculation with serviceability limit states are reported. Such values are directly available for structural elements basing on classification rules in "grading categories" (here, category I, II or III) defined in Table 1 of the standard.

The UNI ISO 11138:2004 [18] provides general criteria that guide all operations of assessment, evaluation of the conservation state and design of interventions, as well as their execution. Here, the phases of the assessment, including the desk survey (step 1), preliminary survey (step 2), decay and check survey (step 3), and structural analysis (step 4), are briefly described. The standard gives special attention to the criteria for the choice and design of interventions.

The Swiss SIA 269/5:2011 [20] suggests an approach valid for all timber elements (structural timber, round timber, wooden products, glued laminated timber, etc.) in existing structures. The first important parameters to be taken into account in step 1 are the influence of the load history and changes in the structure. The standard focuses on step 2, step 3, and step 4 as crucial documentation phases. In fact, the standard underlines the importance of pre-knowledge on the structure, for a suitable realistic representation of it in the static model. Special attention is given to the assessment of connections: direction and kind of transmission of forces, flexibility, eccentricity in the transmission of forces, and finally to wooden nails and dowels, to which Annex A is dedicated. The standard proposes an assessment procedure divided into "knowledge levels" with different depths of knowledge (limited, normal, complete). Furthermore, for step 4, it refers to [31,34,35] and it offers some formulas for the definition of load-carrying capacity. Regarding step 6, the standard gives no advices about technical solutions or interventions' principles, but it refers to SIA 269:2011 [31] and SIA 265:2012 [36] for the calculation in case of addition of new structural elements in the preexisting structure, or changes in the structural system.

The German national annex to the Eurocode 5 [21] gives quantitative information about step 4, how to calculate the evidence for some of the most common woodworking connections (notched connections, single and double step joint, mortise and tenon), as well as the evaluation of behavior of wooden mechanical fasteners (wooden nails and dowels).

*3.5. Proposal for the New European Technical Rules for Assessment and Retrofitting of Existing Structures*

The document on rules for the assessment and retrofitting of existing structures [22] records and lists the current European standards for the assessment of existing structures (Part II), and summarizes them in a harmonic document (Part III). The report presents a guidance applicable for any existing construction and is not material specific. For specific material rules, the document is referring to the specific material Eurocodes. Furthermore, it bases on the principles of structural reliability and consequence of failure, in agreement with the principles of EN 1990:2002 [37].

The proposed analysis focuses on Part III that contains the resume of the most important features contained in the European documents about the assessment process. The generic assessment process is proposed in form of flow-charts. Here, it is inferred that the scope of the assessment is to verify the structural safety and reliability of a structure with respect to its specified remaining working life. The aim of the assessment is the evidence of structural performance/reliability. The document covers all the phases of the assessment procedure in a qualitative form, and for some of the steps more quantitative pieces of information are given. The report proposes the differentiation into different "levels of knowledge" for the assessment, with different depth (limited, normal, complete) based on a "two-step" investigation (preliminary and detailed investigation) that corresponds to step 1 to step 4 of the proposed division. In step 4, a brief description of the content of the structural analysis and verification is given with reference to EN 1990:2002 [37]. Furthermore, for the reliability assessment the ISO 13822:2010 [19], ISO 2394:2015 [28] and the Probabilistic Model Code [13] are referenced. In Section 1, it is highlighted that the report can be also used for the assessment of historical structures, providing additional considerations (step 5) concerning the conservation of the construction identity and authenticity through the conservation of its appearance and materials, but with the specification that the document does "not cover assessment related to special performance requirements of the client related to property protection". Finally, for step 6, general recommended measures and concept of retrofitting through repair or upgrading are reported. Survey and monitoring maintenance are also suggested. For a quantitative design of interventions, Eurocode 2 to Eurocode 9 are referenced.

*3.6. EN 17121:2019*

In the scope of European Cooperation in Science and Technology (COST), two main groups produced documents on the assessment of historical wooden structures: The COST IE0601–WoodCultHer Action and the COST Action FP1101.

The COST IE0601–WoodCultHer Action agreed to produce guidelines and standard documents with the aim of (among others) inspection, assessment, and development of principles and possible approaches for timber grading, safety assessment and criteria for conservation and reinforcement of historical timber structures. Finally, to contribute to European Standardization in the field and produce inputs for CEN/TC 346 Committee "Conservation of Cultural Property". The will to contribute to European standardization was pursued with the awareness that since in the field of cultural heritage each timber artwork is different (materials, wood species, manufacture, history, environment(s), decay/deterioration, interventions, etc.), each artwork needs/deserves a personal care, i.e., individual assessment, evaluation, solutions. Therefore, the developed technical standards pursued the aim of developments of methods and criteria, not standard solutions to problems.

The COST Action FP1101 focused on the production of documents and guidelines regarding the topics of assessment, reinforcement, and monitoring of both historical and engineered timber structures, with the aim to disseminate knowledge for experts in the field. Further aims were to improve the maintenance of existing timber structures and retrieve them for future use, to disseminate up-to-date results to the industry, code writers, policy makers and society for a better interdisciplinary cooperation in the preservation of existing timber buildings. The final report of the action is divided in three main documents relative to the results of three working groups on the assessment of timber structures (WG 1), repair of timber structures (WG 2), and monitoring of timber structures (WG 3).

From all these efforts, a general document from the CEN working group CEN/TC346/WG 10, entitled *Guidelines for Conservation of cultural heritage—Historic Timber Structures—Guidelines for the On Site Assessment* [23], was developed and finally approved as official European document EN 17121:2019 in December 2019.

The standard EN 17121:2019 [23] is based on Cruz, Yeomans, et al. [5] and refers to ISO 13822:2010 [19]. The main purpose of the document is to develop diagnostic methods and criteria specific for the assessment of load-bearing timber heritage structures. The standard can be applied to any kind of timber member, except structural members made of engineered wood, such as glued laminated timber. The document gives a complete qualitative description of the assessment process, except for step 5. Step 1 and step 2 are brought together in a "preliminary assessment" and step 3 and step 4 are named as "detailed survey". Step 2 and step 3 contain quantitative pieces of information: Step 2 gives a brief description of tools for visual NDT, sDT and DT methods, while in step 3, special importance is given to the assessment of structural joints, in terms of both the determination of geometry, alteration, and load-carrying capacity, with a description of their common failure modes for load cases. For step 3, some basic principles of timber grading are listed, referencing EN 14081–1:2016, Annex A [38]. Moreover, for the determination of moisture content, the EN 13183–3:2005–06 [39] is provided as a reference. As already mentioned, the document does not provide any special information about step 5 for special requirements connected to heritage structures, and step 6 is limited to a qualitative description of the design of interventions based on the principles of minimal intervention, incremental approach, and removable measures.

## 4. Conclusions

From the comparison among literature papers, guidelines, codes, and standards, interesting results on European knowledge, and interests in the assessment of existing timber structures emerged.

From the reviewed literature, it emerged that all steps for a correct assessment project are covered and described in existing scientific papers and articles. However, the analyzed literature does not show a unanimous definition of the "terms" of the assessment procedure. The main topics discussed are the assessment of the condition state of the structure (step 2 and step 3), and general qualitative advice on how to perform a structural analysis (step 4). These steps assume, from author to author, different levels of in-depth analysis.

Existing European guidelines, codes and standards also reflect the same situation stated in the literature. Documents that debate the assessment of timber heritage structures are in smaller number compared to the ones that debate on the assessment of the general heritage structural material, and their contents are not unanimous in the definition of the "terms" of the assessment procedure. However, the *New European Technical Rules for the assessment and retrofitting of Existing Structures* [22] constitutes a starting point for the assessment of the general (heritage) structural material. Nevertheless, further efforts are necessary to implement such a document for each of the specific construction materials, including for timber.

In general, the less debated topics both in the literature and European standards are the special requirements (step 5) and design of interventions (step 6). This latter topic has been mostly covered for modern timber structures, but less for heritage timber structures. The reason for this could lay in the complexity of the solution-finding for a single heritage structure. The reason of the deficiency in step 5 can lay in the complexity and variety of local laws, and the consequent difficulty to harmonize them in a common European norm. The consequence is that the developed international standards and guidelines do not offer sufficient technical information to be of any practical significance during these two latter steps of the assessment procedure. However, the tools for the assessment of timber structures, both praxis and normative tools [20,21], are acknowledged at the European level and constitute the foundation for the development of a more homogeneous process of assessment.

In conclusion, the present paper offers a first approach to the harmonization of the process of structural assessment of heritage timber structures at European level, and to the development of

guidelines for the preservation of timber heritage structures. This kind of conclusion is also valid for the assessment of structures of other historic materials. Moreover, this paper offers a homogenization of the "terms" and "definitions" of the structural assessment, at least for step 1 to step 4 of the proposed assessment process. Nevertheless, this paper limits the definition of the process of structural assessment to structural aspects of heritage timber structures, while historical-aesthetic aspects, which may be relevant for the first steps of the proposed methodology, are not the focus of the discussion. At the same time, this paper does not address the issue of seismic vulnerability with the necessary level of detail. In conclusion, these two latter aspects must be considered in future research on assessment methodologies for timber heritage structures.

In conclusion, this paper constitutes a further milestone for the achievement of an exhaustive European harmonic standard in the preservation of heritage timber structures. Furthermore, this paper strives to complement the European Standard ISO 13822:2010 [19] or the New European Technical Rules [22] for the assessment of timber heritage structures. This would be of basic importance because both standards will be able to ensure, on one side the correct preservation methodology for the built heritage of any construction material, and on the other side, an international, interdisciplinary dialogue between experts whereby a better common understanding of the cultural heritage can be achieved.

**Author Contributions:** Conceptualization, E.P. and M.S.; Data curation, E.P.; Formal analysis, M.S.; Investigation, E.P.; Methodology, E.P. and M.S.; Project administration, E.P.; Supervision, M.S.; Validation, M.S.; Writing—original draft, E.P.; Writing—review and editing, E.P. All authors have read and agreed to the published version of the manuscript.

**Funding:** This research received no external funding.

**Conflicts of Interest:** The authors declare no conflict of interest.

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
