# Peer review of "Six-Steps Process of Structural Assessment of Heritage Timber Structures: Definition Based on the State of the Art"

_buildings, doi:10.3390/buildings10060109_

Round 1

Reviewer 1 Report

An interesting article summarizing the guidelines contained in the literature and in codes or norms and an attempt to systematize them.
Thank you for taking this difficult task. Wood is a specific material whose properties depend on many variables, including the regions from which they come and the regulations that apply. Therefore, it is important to evaluate this material also in a local aspect.

The published literature is rich and current, but limited to some regions, it is recommended to extend the sources.

It is recommended to improve the readability of Table 1 - its meaning is difficult to understand.

Verse 149
- lack of legibility STEP 3: Decay & Check Survey
- lack of legibility STEP 5: Special

Tables 2a and 2b - illegible (truncated) descriptions in column 5

Author Response

Thank you to appreciate my research.

The tables and figures were edited and their readability is improved.

An extension of the sources need more research from my side and could be the topic for another paper that cover the standards in extra-european regions.

Reviewer 2 Report

The author has analysed papers, reports, guidelines and norms on the assessment of historic timber structures and makes tables to indicate which topics are discussed by which authors.

Apparently, a global document that discusses the steps 1 to 6 is missing today, although the steps are discussed separately in various publications. Probably, this kind of conclusion is also valid for the assessment of many other historic materials.

The actual text (in its current form) reads as a state-of-the-art that is written before the research goal of a project is explained. It is a valuable and scientific overview of the existing literature, standards and norms in some European countries, but is the information valuable enough to publish it as an independent paper?

If the paper is accepted for publication, the introduction could be rewritten: the idea that is now developed in the conclusion could be taken as the starting point of the paper: As a conclusion, the first step for the harmonization of the process of structural assessment of timber structures at European level is the homogenization of the “terms” and “definitions” of the structural assessment at least for Step 1 to Step 4 of the proposed assessment process. “

Author Response

Thank you to appreciate my research.

I add the reflection that this kind of discussion is important also for other construction materials in the conclusions.

The introduction was rewritten according to your suggestions.

Reviewer 3 Report

The paper deals with an elaborate analysis of existing timber structure assessment methodologies. The main scope of the paper is to bring forward the main approaches of each analysed methodology and highlighting the main differences between them in order to be able to develop future research directions.

Divided into two sections, the paper is first presenting a review of papers written, which describe various assessment methodologies suitable for both historical and contemporary timber structures. Fifteen different approaches are analysed in this section, containing papers written between 2000 and 2018 all around Europe. 6 main steps considered for the assessment of historic structures are identified and three milestones. Each paper is reviewed considering the identified steps and milestones by analysing the content of each assessment methodology and highlighting the differences between the considered steps. A comprehensive table is also presented, which contains a summary analysis of the considered steps in each study.

In order to better explain the content of each step and the main objective of each milestone, a figure is also included in the paper, which is clearly presenting the main findings of the reviewed papers.

In the second section, the focus is shifted towards the review of existing codes and guidelines which are trying to define a proper assessment approach suitable for timber structures. The review is presenting some of the most important codes, from ICOMOS charters to various national standards (Italian, Swiss, German) and European standards. Again, in order to transmit the differences between the approaches more clearly, a comprehensive table is offered, containing information regarding the way each methodology is approaching the six steps. At the same time, the table is also including additional information concerning where data about each step can be found in the codes and standards.

After the comprehensive review of both papers and codes, the main conclusions are brought forward, highlighting that the topics considered in each methodology are approximately similar.  However, there are still certain stages of the assessment, which are still not adequately analysed, like the special requirements step or the design of interventions step. It is also highlighted that there are still no common definitions concerning the considered terms of each assessment methodology which is preventing a clear dialogue between various experts.

The presented review is very interesting and approaching a topic which is up to date since clear guidelines have to be developed, which can ensure the safety of the built heritage. At the same by considering this state of the art of timber heritage structure assessment-related papers and codes, the authors highlight that without a proper understanding of the strengths of each previously developed methodology and identification of the missing topics it is not possible to understand valuable heritage structures properly. The performed review is thoroughly done and is clearly presenting the main features of each analysed methodology. By highlighting the presence of the six steps and three milestones and how they are considered in each methodology, the main differences and similarities between the 15 considered papers and 11 analysed codes and standards are therefore brought forward.

There are still some minor revisions necessary:

Figure 1 – there seems to be an error in the provided figure since the name of each step is only partially visible (Step 3 and step 5) or completely invisible (Step 6)

Page 7 – line 190-191 – the sentence is rather difficult to understand and is written in a completely different style than the rest of the paper. Please consider revising this sentence

Considering the complexity of the performed review, the conclusions are not highlighting a problem which also emerges from the performed analysis. Although in the conclusions the authors are highlighting that an interdisciplinary dialogue is necessary in order to ensure the safety of timber heritage structures, the performed review highlights that the assessment methodologies are mainly focused on their structural features. They are not considering data about the context of the timber heritage building, its aesthetics or even link to the building if the assessment is focused on timber roof structure, features which also have to be considered in the first steps of the methodologies. The analysis of all these features could offer additional information about the structure and its value and subsequent vulnerability. It is therefore recommended also to highlight that future methodologies also need to consider the complexity of these structures and explain how a multi- and interdisciplinary approach can be ensured.

At the same time, since the seismic vulnerability of timber roofs is mentioned in the paper, the effect of the presence of certain timber elements or even the roof structure on the behaviour of masonry buildings in seismic areas should also be mentioned, as an criterion which should be taken into consideration in future assessment methodologies.

Author Response

Thank you for appreciating my research.

Minor revisions were improved.

The reflection about the importance of historical-architectonical-easthetic aspects and necessity of an in-deepended research on such topics is added in the conclusions.

Round 2

Reviewer 2 Report

The remarks are well integrated in the text.